# Effects of Reactive Oxygen Levels on Chilling Injury and Storability in 21 Apricot Varieties from Different Production Areas in China

**DOI:** 10.3390/foods12122378

**Published:** 2023-06-15

**Authors:** Qi Xin, Xinqun Zhou, Weibo Jiang, Min Zhang, Jing Sun, Kuanbo Cui, Yu Liu, Wenxiao Jiao, Handong Zhao, Bangdi Liu

**Affiliations:** 1Academy of Agricultural Planning and Engineering, Ministry of Agriculture and Rural Affairs, Beijing 100125, Chinaxinqunzhou@163.com (X.Z.); zhangmin7060@163.com (M.Z.); sunjing@appe.org.cn (J.S.); jiagongsuo1@163.com (Y.L.); 2Key Laboratory of Agro-Products Primary Processing, Ministry of Agriculture and Rural Affairs of China, Beijing 100125, China; jwb@cau.edu.cn (W.J.); xin147258369092985@163.com (K.C.); jiagongsuo2@163.com (W.J.); jiagongsuo3@163.com (H.Z.); 3College of Food Science and Nutritional Engineering, China Agricultural University, Beijing 100083, China; 4Institute of Agricultural Mechanization, Xinjiang Agricultural University, Wulumuqi 830091, China; 5School of Food Science and Engineering, Qilu University of Technology, Jinan 250353, China

**Keywords:** apricot, varietal differences, chilling injury, chilling tolerance during storage, reactive oxygen species metabolism

## Abstract

The key factors for resistance to chilling injury in apricot fruits were obtained by analyzing the low-temperature storage characteristics of 21 varieties of apricot fruits in the main producing areas of China. Twenty-one varieties of apricots from different production areas in China were stored at 0 °C for 50 d and then shelved at 25 °C. The storage quality, chilling injury, reactive oxygen species (ROS), antioxidant ability, and contents of bioactive substances of the apricots were measured and analyzed. The results showed that the 21 varieties of apricot fruits could be divided into two categories according to tolerance during low-temperature storage, where there was chilling tolerance and lack of chilling tolerance. Eleven varieties of apricots, of which *Xiangbai* and *Yunbai* are representative, suffered from severe chilling injury after cold storage and shelf life. After 50 d of storage at 0 °C, the levels of superoxide anions and hydrogen peroxide accumulated in the 11 varieties of apricots with a lack of chilling tolerance during storage were significantly higher than those in the remaining 10 varieties of apricots with chilling tolerance. In addition, the activities of ROS scavenging enzymes, represented by superoxide dismutase, catalase and peroxidase, were significantly decreased in 11 varieties of apricots with a lack of chilling tolerance during storage. The contents of bioactive substances with ROS scavenging ability, represented by ascorbic acid, total phenols, carotenoids, and total flavonoids, also significantly decreased. The 10 varieties of apricots, of which *Akeximixi* and *Suanmao* are representative, were less affected by chilling injury because the production and removal of ROS were maintained at normal levels, avoiding the damaging effects of ROS accumulation in the fruit. In addition, the 10 apricot varieties with chilling tolerance during storage had higher sugar and acid contents after harvest. This could supply energy for physiological metabolism during cold storage and provide carbon skeletons for secondary metabolism, thus enhancing the chilling tolerance of the fruits. Based on the results of cluster analysis combined with the geographical distribution of the 21 fruit varieties, it was found that apricot varieties with chilling tolerance during storage were all from the northwestern region of China where diurnal temperature differences and rapid climate changes occur. In conclusion, maintaining the balance of ROS production and removal in apricots during cold storage is a key factor to enhance the storage tolerance of apricots. Moreover, apricots with higher initial glycolic acid and bioactive substance contents are less susceptible to chilling injury.

## 1. Introduction

Apricot is a drupe that is extensively grown and consumed worldwide. It is widely cultivated in various regions and areas of the world, including the Americas, Asia, the Mediterranean, and South Africa. China has 90% of the world’s apricot germplasm resources, and the northern Chinese provinces of Xinjiang, Gansu, Hebei, Shaanxi, Shanxi, and Henan are considered to represent regions where apricot cultivation originated [1]. Although rich in germplasm resources, the apricot consumption and processing industry in China has gradually shrunk in recent years mainly due to the lack of storage and preservation techniques, resulting in stagnant consumption. There are 10 macrospecies, 13 species and more than 2000 varieties of apricots. The tolerance of different apricots to cold storage varies greatly, but there is a lack of exhaustive research to support the preservation and storage techniques of different varieties of apricots [2]. Therefore, it would be valuable to study the variability in the storage of apricots.

Apricots are produced and harvested in China from May to August each year. As apricots are a typical respiratory climacteric fruit, they are mainly stored at low temperatures from 0 to 4 °C in combination with other preservation treatments, with a storage and marketing period of 40 to 50 days. Stanley et al. [3] used 0 °C storage of Larclyd apricots to extend the storage cycle to 40 days, but the study suggested that physiological disorders occur in apricots after prolonged storage. Liu et al. [4] used a novel near-freezing temperature technique for storing Xingmei apricots and found that although it was effective in extending the storage period to 60 days, severe chilling injury occurred during the 60-day post-shelf life period.

The specific fruit variety involved is an important influencing factor that affects most postharvest fruit and vegetable chilling injuries. Because the nutrient and bioactive substance contents of different varieties of fruits and vegetables vary greatly, their sensitivities to low temperatures are also different. In addition, varieties of the same species may behave differently in different geographical areas under similar temperature conditions [5]. In a study of postharvest chilling tolerance of 23 melon varieties in Chile [6], three varieties ‘Hy-Mark’, ‘Gal 96’, and ‘Voyager I’ had the highest chilling tolerance and could be preserved for longer periods. Carvaja et al. [7] evaluated the postharvest chilling tolerance of several different varieties of zucchini and found that the ‘Natura’ variety was best suited for storage at 4 °C. Studies concerning the storage of different grape varieties have also found that “red-skinned” grapes have generally higher chilling tolerance and have a longer storage life than “yellow-skinned” grapes [8]. In addition, different varieties of fruits and vegetables of the same genus usually have different places of origin and harvesting seasons. Most scholars believe that the place of origin is also a reason that affects the chilling tolerance of fruits and vegetables [9]. Fruits native to temperate zones have higher chilling tolerance than those native to the subtropics and tropics and have longer storage periods [10]. At the same place of origin, fruits ripening in autumn and winter have higher chilling tolerance than those in spring and summer. In a study on the storage of kiwifruit of different varieties and places of origin, it was found that the variety “Xuxiang”, which reaches full maturity at high latitudes, had better chilling tolerance [11,12].

In addition, the regulation of reactive oxygen species (ROS) levels is considered to be a key regulator of resistance to chilling injury in fruits and vegetables. When postharvest fruits and vegetables are subjected to cold stress, ROS are produced and accumulated. This causes oxidation chain reactions that lead to oxidative damage to cell membranes and loss of cell membrane function [13]. In a study by Chen [14], glycine betaine was used to treat bananas. Results showed that glycine betaine could not only enhance the contents of antioxidant substances but also reduce Chilling Injury(CI) symptoms in banana fruit. Liu [15] used near-freezing temperature (NFT) storage of apricots and found that compared with 0 °C storage, NFT storage effectively maintained Superoxide Dismutase (SOD), Aseorbateperoxidase (APX), and Catalase (CAT) activity and antioxidant active substance content, antioxidant capacity, and reduced malondialdehyde and hydrogen peroxide accumulation during storage [16]. Apricots were protected from normal initiation of ethylene metabolism during shelf life and alleviating CI of inability to ripen.

In this study, 21 varieties of apricots from northern China were subjected to cold storage and post-storage shelf life experiments to investigate the occurrence of chilling injury and changes in ROS metabolism, and to investigate the relationship between ROS metabolism and chilling tolerance in different varieties of apricots in order to lay a foundation for the research of storage and transportation techniques of apricots. This study could provide a theoretical basis for the study of storage and shelf life techniques for different apricot varieties.

## 2. Materials and Methods

### 2.1. Acquisition of Materials of 21 Varieties of Apricots

All 21 varieties of apricots used in the trial were harvested in the same year (2022), and the 21 varieties were produced in 11 provinces in the northern region of China. The specific harvesting times, variety names, and their abbreviations are shown in Table 1. All apricots were harvested at the same state of maturity, 120 ± 2 days after flowering, and all apricots were harvested with <20% surface discoloration [17]. Due to the large variation in fruit size, each variety of apricot was guaranteed to have at least 500 fruits of similar shape, color, and size for experiments.

Apricots from different origin regions were picked directly by local farmers and transported to the laboratory in Beijing via a 10 ± 2 °C cold chain transport truck and aircraft. All apricot samples were transported from the origin regions to the laboratory in Beijing within 48 h via 10 ± 2 °C refrigerated trucks and aircraft.

### 2.2. Apricot Cold Storage and Shelf Treatment Methods

After arriving at the laboratory, the apricots of each variety were transferred to cold storage, and they were subjected to a process of standing at 10 °C for 8 h. Apricot fruit samples that have completed the standing process would be washed with deionized water and natural air dried.

The washed fruits without damage, disease or obvious abnormalities were selected for storage and shelf-life experiments. The washed apricots were snap-frozen in liquid nitrogen and left as origin samples; 75 kg of apricots of each variety were divided into three parallel groups of 25 kg and placed in plastic baskets with perforated, 40 μm-thick polyethylene bags. Cold storage was carried out at 0 ± 1 °C. Observations were made every 15 days for the occurrence of evident chilling injury and disease. After 50 days of storage, 20 kg of each apricot variety was frozen in liquid nitrogen and retained as “storage samples”. In addition, 5 kg of fruits without chilling injury and decay were selected and placed at room temperature (25 ± 2 °C) for shelf life experiments. When the shelf life experiment was completed, the apricot fruit would be frozen in liquid nitrogen as a “shelf life sample”.

### 2.3. Color, Firmness and Weight Loss Rate Determination

Fruit flesh firmness was determined by measuring opposite peeled sides of the fruit using a texture analyzer (GY-3, Tuopu Instrument Co., Ltd., Ningbo, China) with a 3.5 mm probe and expressed as Newtons (N).

The weight of single fruit was measured by precision electronic analytical balance (±0.0001 g), and the data were recorded. In each storage and shelf life group, 20 fruits were selected for marking, and the weight loss rate was calculated before and after storage.
$$Weight\;loss\;Rate=\frac{marked\;fruit\;initial\;weight-marked\;fruit\;weight\;after\;storage}{marked\;fruit\;initial\;weight}\times 100\%$$

The color parameters *L**, *a** and *b** of different varieties of apricot pulp were measured by colorimeter (NR110, 3NH Technologies Inc., Shanghai, China), and the total color difference Δ*E* was recorded and calculated (the formula is as follows). Photos of all varieties of origin fruits, naturally ripened fruits, post-storage fruits, and post-storage shelf samples were taken under the same dark room black background and lighting, and the same camera was used for shooting to ensure the most realistic picture performance.
$$\Delta E=\sqrt{(L^{*}-L_{0}){}^{2}+(a^{*}-a_{0}){}^{2}+(b^{*}-b_{0}){}^{2}}$$

### 2.4. Determination of Soluble Sugar and Organic Acid Content, Ascorbic Acid, Total Carotenoids, Total Flavonoid Content and Total Phenolic Content

The determination method of total carotenoid (TC) referred to the colorimetric method [18] and was modified. We weighed 10 g of apricot pulp ground in liquid nitrogen, placed it in a mortar, added 100 mL of hexane/acetone/ethanol organic extractant (volume ratio: 50/25/25), ground and mixed well, transferred it to a conical flask, and shook it in a shaker for 60 min. We added 20 mL distilled water into the conical flask and shook it well, transferred it to the liquid separation cone, took the supernatant and kept it for use, we used the nitrogen blower to dry it in cold air, then used a tetrahydrofuran/acetonitrile/methanol dissolver (volume ratio: 15:30:55) to fully dissolve the blow-dried residue and fix the volume to 20 mL, which is the total carotenoid extraction solution, stored under freezing for use. The absorbance was measured at 446 nm by spectrophotometer (L5S, Shanghai, China), and the content of TC in apricot fruit was calculated by using hexane-dissolved β-carotene as the standard equivalent. The unit was g·kg^−1^.

Ascorbic acid content, total flavonoid content (TFC), and total phenolics content (TPC), in the apricot flesh were determined as previously reported [19]. TFC was expressed as mg of rutin equivalents (mg·g^−1^) per sample mass, based on the standard curve obtained from the gallic acid standard. TPC was expressed as mg of gallic acid equivalents (mg·g^−1^) per sample mass, based on the standard curve obtained from the gallic acid standard.

The soluble sugar content (SSC) was measured by a hand-held sugar meter (PaL-BXIACID F5, Tokyo, Japan). About 30.0 g of apricot pulp was stirred in a tissue stirrer to a pulpy state, and a clear pulpy juice was obtained by squeezing and filtering with a multi-layer gauze. The drip was sucked out on the sugar meter and the SSC was measured and recorded. The experiment was repeated three times.

Organic acids were determined by acid-base neutralization titration. Referring to the improved method from Ma et al. [20,21]; 2–5 g of pulp ground in liquid nitrogen (adjust the pulp weight according to different varieties) was weighed. After adding 5 mL of distilled water, the pulp was homogenized and transferred to a 10 mL centrifuge tube. After centrifugation at 4 °C, 10,000 r·min^−1^, the supernatant was taken for use; 2 mL of the supernatant was added to a 50 mL conical flask, 20 mL of deionized water was added, and three drops of 1.0% phenolphthalein were added as the titration indicator. The liquid in the conical flask was titrated with the calibrated 0.01 mol·L^−1^ NaOH solution to a stable pink color, and the volume of the NaOH solution consumed was recorded. The titratable acid of apricot was calculated with malic acid as the reference standard (conversion coefficient = 0.067). The detailed calculation formula is shown in Cao et al. [22].

### 2.5. Statistics of Chilling Injury Rate and Chilling Injury Index

In our study, the major chilling injury symptom in the 21 varieties of apricots was found to be flesh flocculates, browning, water-soaking lesions and peel-pitting spots [15]. Thus, after storage and shelf life, 40 apricots were randomly selected for the statistics on the incidence of chilling injury.
$$CI=\frac{Number\;of\;chilling\;injury\;fruits}{Total\;number\;of\;fruits}\times 100\%$$

The development of the chilling injury index (CII) was subjectively assessed using 40 fruit from each of three replicates according to a previously described method [23] with minor modifications. The degree of flesh flocculates, internal browning, watery injury and peel pitting spot were estimated by rating apricots on a scale of 0–4 as follows: 0 = none CI, 1 = slight (0 < chilling injury < 1/4 total area), 2 = moderate (1/4 total area < chilling injury < 1/3 total area), 3 = moderately severe (1/3 total area < chilling injury < 1/2 total area), and 4 = severe (1/2 < chilling injury).
$$CII=\sum \frac{Score\times number\;of\;fruits\;at\;the\;corresponding\;score\;level}{The\;highest\;chilling\;injury\;score\times Total\;number\;of\;fruits}$$

### 2.6. Determination of MDA Content and Cell Membrane Permeability

The 30 flesh disks (5 mm diameter × 5 mm thickness) of the ten fruit were used to determine the ion leakage [4]. The disks were steeped in doubly distilled water (50 mL) in glass vials for 60 min, and the solution conductivity was tested with a conductivity apparatus (DDS-11A, Shanghai INESA Scientific Instruments Co., Ltd., Shanghai, China). Then the disks were boiled in a water bath for 30 min and cooled to room temperature, so the total conductivity was collected. Ion leakage was displayed as relative conductivity.
$$MDA=\frac{conductivity\;of\;tissue\;solution}{total\;conductivity}\times 100\%$$

The malondialdehyde concentration (MDA) was analyzed as reported previously [4] with some modifications. Five grams of fresh apricot tissues were homogenized with 5 mL 30 mmol·L^−1^ trichloroacetic acid and then centrifuged at 10,000× *g* for 10 min at 4 °C. The concentration of MDA was expressed as mmol·g^−1^ fresh weight.

### 2.7. Determination of Hydrogen Peroxide, Superoxide Anion Production Ability and DPPH Free Radical Scavenging Ability

Hydrogen peroxide (H_2_O_2_) content was tested according to [24,25] with some modifications. Apricot flesh tissue (10 g) was homogenized in 5 mL of cold acetone and centrifuged at 10,000× *g* and 4 °C for 30 min. About 1 mL supernatant was mixed with 0.1 mL of 22 mmol L^−1^ titanium sulfate and 0.2 mL ammonia, and the mixture was centrifuged at 10,000× *g* and 4 °C for 15 min. Then, the pellets were dissolved in 3 mL of 1 mol·L^−1^ sulfuric acid and centrifuged for 15 min at 10,000× *g*. H_2_O_2_ production was tested using H_2_O_2_ as a standard and expressed on a fresh weight basis as mmol g^−1^.

The determination of 1,1-dipheny1-2-Picryl-Hydrazyl (DPPH) clearance capacity referred to Xin et al. [25] and was modified; 5 g of frozen callus were taken and placed in a centrifuge tube (50 mL). The 20 mL ethanol (70%) solution was added, mixed, and ultrasonically treated for 1 h, centrifuged (4 °C 10,000× *g* 20 min), and the supernatant was taken as the extract. The 0.05 mL extract was absorbed and placed in a test tube and 3 mL DPPH ethanol solution with a concentration of 0.3 mmol·L^−1^ was first added and mixed. After a water bath at 30 °C for 1 h, the reaction was immediately cooled and terminated. DPPH radical scavenging capacity was calculated using Trolox as the standard equivalent, expressed in mg·g^−1^.

The production capacity of superoxide anion was determined by the Nanjing Jiancheng assay kit [26]; 2 g tissue was weighed and placed in a 10 mL centrifuge tube, and 5 mL 0.05 mol·L^−1^ 7.8 pH phosphate buffer (containing 0.001 mol·L^−1^ ethylenediaminetetraacetic acid, 0.3% Triton X-100 and 2% polyvinylpyrrolidone) was added. After mixing, the supernatant was collected by centrifugation (4 °C 10,000 r·min^−1^ 20 min). According to the instructions of the determination kit, the corresponding reagents were added in turn. After mixing, the absorbance value was measured at 550 nm. O^2−^ content was expressed as nmol·min^−1^·g^−1^.

### 2.8. Activity of Antioxidant Enzymes

Superoxide dismutase (SOD) activity was determined using the method described by Wang et al. [27]. Briefly, 0.1 mL sample extract A was mixed with phosphate buffer (50 mM, pH7.8), 75 µM nitro-blue tetrazolium (NBT19), 10 µM EDTA20, 2 µM riboflavin and 13 mM methionine. The reaction mixture was incubated for 20 min under fluorescent light and read for absorbance at 560 nm and converted into an enzyme activity Unit (U). One unit (U) was defined as inhibition of NBT photo-reduction by 50%. SOD activity was expressed as U.

Catalase (CAT) activity was assayed based on the method described with some modifications [28]. Briefly, 0.3 mL extract A was mixed with 0.6 mL H_2_O_2_ (50 mM) and 2.1 mL phosphate buffer (50 mM, pH7.0), followed by incubation at 30 °C for 1 min and the absorbance was measured at 234 nm. The CAT activity was expressed as U·g^−1^, and one unit (U) of catalase was equivalent to the conversion of 1 µmol H_2_O_2_ per minute.

The method of Wang et al. [27] was used to determine the Peroxidase(POD) activity. Briefly, 0.3 mL extract A was mixed with 2 mL of 0.1% guaiacol and 1 ml of 0.1 mM H_2_O_2_ and reacted at 30 °C for 5 min. The absorbance was measured at 460 nm for the POD-specific activity that was expressed as U. One unit (U) refers to the amount of enzyme required to increase the absorbance by 0.001 per minute.

### 2.9. Statistical Analysis

Three biological replicates were performed in all experiments, and three technical replicates were performed during the index determination. The experimental results were expressed as mean ± standard deviation. Duncan’s multiple comparison method of IBM SPSS Statistics 19 software was used to analyze the differences between data (*p* < 0.05), and HIPLOT 0.2.0 was used for cluster heat map analysis (the significance of different types of values is marked and distinguished by lowercase letters).

## 3. Results

### 3.1. Quality Changes and Shelf-Life Ripening Performance after Cold Storage of Different Varieties of Apricots

Quality changes to the 21 apricot varieties after storage at 0 °C for 50 d and after shelf life at room temperature are shown in Figure 1. Except for YBI, the remaining 20 apricot varieties could be effectively stored until day 50. Figure 1 shows the visual photos of 21 kinds of apricot fruits in their initial state (*a*), natural ripening, refrigerated and shelf state. All apricots had significant color changes in terms of both skin and flesh after prolonged storage. As shown in Figure 1, The color change to apricots occurred mainly during the shelf life period, and flesh color change was more evident than that of skin color; therefore, the study focused on pulp color change. The apricot flesh had two main color changes. The flesh color of 12 apricot varieties, of which AKI, BTU, SSN, and ZZU are representative, changed from white and light yellow to orange, while the flesh color of nine apricots varieties, of which BSI, DLI, HMI, and SMO are representative, changed mainly to bright yellow. From Figure 2B, it can be seen that the 21 apricot varieties with the largest flesh color change during storage were YBI, CZH and AKI. The large difference in color difference values for YBI was due to severe browning of the flesh, while the flesh color of both CZH and AKI changed to a darker orange color after 50 days of storage. During storage, the flesh color changes were minimal for SMO and DLI. Some of the apricots had a significant color change after shelf life due to post-ripening, mainly XBI, SJG and SSN.

Softening was also evident in apricots after storage and shelf life, as shown in Figure 2A. Twenty-one apricot varieties were softened by almost 20% after prolonged cold storage, with only BSI, BTU, SMI, and SMO maintaining high levels of firmness after storage. Additionally, after shelf life, all apricots showed varying degrees of softening. BSI and SMI, which maintained high levels of firmness during storage, exhibited significant softening after shelf life. The only fruit that still maintained a high degree of firmness (<40% softening) after shelf life was SMO. The three fruits with the highest degree of softening after shelf life were DJE, CZG and ZZU, all with softening rates greater than 80%.

As can be seen in Figure 2C, none of the 21 apricot varieties had a weight loss rate of >5% in storage and after shelf life. All the varieties except for two, KCI and LTI, showed <3% weight loss after storage and shelf life. Figure 3 shows the changes in sugar and acid contents of different varieties of apricots after storage and shelf life. The soluble sugar content of apricots after storage was higher than the initial samples for all 18 varieties except YBI, JTG and DJE. By combining Figure 1 and Figure 3A, it was found that the decrease in soluble sugar content of YBI, JTG and DJE apricots after storage could be related to chilling injury, and all three apricots showed chilling injury evident to the naked eye after storage and shelf life. All 21 apricot varieties showed a gradual decrease in organic acid content from initial to post-storage and shelf life. Except for SMO and JTG, the organic acid content of the 19 varieties was significantly lower (*p* < 0.05) than the initial state after storage. The organic acid content of all varieties after shelf life decreased to a level significantly different from their initial state *(p* < 0.05). Among them, SMO showed the greatest decrease in organic acid content after storage and shelf life, but the final organic acid content after shelf life was still the highest among the 21 apricot varieties.

### 3.2. Chilling Injury in Different Apricot Varieties after Cold Storage and Shelf Life

Figure 4 represents the indicators related to the occurrence of chilling injury in apricot post-storage and shelf life. Chilling injury did not occur in all 21 apricot varieties during 50 days of cold storage (Figure 4A). There was no chilling injury evident to the naked eye in AKI, BTU, CZG, DLI, LGG, SMI, SSN, and SMO apricot varieties post-storage and shelf life. The incidence of chilling injury shows that most of the apricot varieties exhibited chilling injury in the shelf-life performance after prolonged cold storage. DJE, HMI, JTG and KTY all had 100% incidence after storage and shelf life, i.e., significant chilling injury was observed in all stored fruits. As can be seen from Figure 1, the most important phenomenon in chilling injury occurring in 21 varieties of apricots was “water staining”, followed by the browning of the flesh observed in JTG and YBI, and punctiform depressions of the epidermis observed in BSI, DJE, SJG. Although chilling injury occurred in 13 of the 21 varieties of apricots after storage and shelf life, not all varieties had extremely pronounced symptoms and indicators of chilling injury. The chilling injury index in Figure 3B was calculated by combining the rate of chilling injury and the level of chilling injury. It can be observed that seven apricot varieties BSI, DJE, HMI, JTG, KTY, YBI, and YU had more severe chilling injury, while six apricot varieties KCI, LSG, LTI, SJG, XBI, and ZZU exhibited chilling injury, but the symptoms and phenomena were not obvious. Only YBI and KTY varieties had chilling injury indices exceeding 5 after storage, whereas the remaining apricot varieties that developed chilling injury had low chilling injury indices during storage. The chilling injury index of apricots post-shelf life, of which BSI, DJE, HMI, JTG, KTY, and YU are representative, steeply increased >5. This indicates that although the 21 varieties of apricots were more susceptible to chilling injury, the observable chilling injury occurred mainly during the shelf life period after long periods of cold storage.

Cell membrane permeability is indicative of the degree of cellular disruption (Figure 4C), and MDA (Figure 4D) is commonly used for characterization of the degree of membrane lipid oxidation, which represents highly relevant data for CI. These two indicators showed that cell membrane permeability and MDA content could initially be detected in the fruit, but both values were low. Both cell membrane permeability and MDA content showed different degrees of increase after prolonged cold storage and shelf life. The YBI variety showed a significant increase in both indicators due to severe chilling injury during storage. Apricot varieties with cell membrane permeability exceeding 80% after shelf life were BSI, DJE, HMI, JTG, KTY, LSG, XBI, YBI, YU and ZZU. Similarly, the MDA content of these 10 apricot varieties mentioned above also exceeded 0.06 mmol·g^−1^, indicating a high degree of overlap between these two values. Meanwhile, the other nine apricot varieties, except for ZZU, also suffered from more severe chilling injuries.

### 3.3. Changes in ROS Metabolism and Bioactive Substances in 21 Apricot Varieties after Cold Storage

Figure 5A,B show the superoxide anion production capacity and cellular hydrogen peroxide accumulation in apricots during storage and shelf life. As can be seen from Figure 5A, the superoxide anion production capacity of the 21 apricot varieties was divided into two main categories. One category was that the ability of the superoxide anion to regenerate gradually decreases, from the initial state to after storage and shelf life. This mainly applied to the following 11 varieties: BSI, CZG, DJE, HMI, JTG, KTY, LSG, SJG, XBI, YBI, YU. The other category was that a peak appears after a prolonged period of cold storage, and rapidly declines after shelf life. This mainly applied to the remaining 10 varieties not mentioned above. It is evident that apricots with progressively lower superoxide anion production capacity were those that were more susceptible to chilling injury after storage. The initial hydrogen peroxide content of all 21 apricot varieties was below 5 mmol·g^−1^, and all apricots had significantly higher hydrogen peroxide content than the initial samples after storage and shelf life (Figure 5B). The hydrogen peroxide content of BSI, YBI, DJE, and other varieties which were susceptible to chilling injury increased significantly after prolonged storage. The hydrogen peroxide content after storage was approximately double that of the initial sample and continued to increase significantly after shelf life. This indicated that apricots with severe CI occurrence may be in a state of significant hydrogen peroxide accumulation during storage.

Figure 5C represents the ability of apricots to scavenge reactive oxygen radicals. In complete contrast to the results for hydrogen peroxide, the DPPH radical scavenging capacity of apricots decreased with the storage period. In addition, the difference in DPPH radical scavenging ability in the initial samples was very evident, and the initial DPPH values of the SMO and LGG varieties were two times higher than those of the BSI and XBI varieties. In the DJE, KTY and YBI varieties, where chilling injury was more severe, post-storage DPPH values decreased significantly. This suggested a greater correlation between the occurrence of chilling injury and the loss of free radical scavenging capacity. The three main enzymes associated with ROS free radical scavenging are shown in Figure 5D–F. Among them, SOD activity showed a significant decrease after prolonged storage and shelf life, while there were signs of POD and CAT activation after storage. The post-shelf life SOD activity of AKI, BTU, LGG, SMO and other varieties that did not exhibit chilling injury and were well stored remained high. The post-shelf life SOD activity of SMO was 79.5% of that of the initial sample. However, the SOD activity of KTY, DJE, HMI and other varieties with severe chilling injury decreased significantly. For example, the post-shelf life SOD activity of KTY was 31.6% of the initial sample. The POD and CAT activities of all 21 apricot varieties showed trends of activation of varying degrees after storage and shelf life. The POD activities of all apricot varieties were lowest in the initial state and the highest during shelf life, and the post-shelf life POD activity of all apricot varieties exceeded 15 U·g^−1^. The trend involving SOD was slightly different from that of POD. CAT activity levels in the 10 apricot varieties of BSI, CZG, DJE, HMI, JTG, KTY, LSG, SJG, YBI, and YU were significantly higher than the post-shelf life CAT activities at the end of the storage cycle. This indicated that the CAT enzymes of these 10 apricot varieties may be stressed by low-temperature environments and show rapid inactivation post-shelf life.

Figure 6 shows the results of testing 21 biologically active plant compounds in apricots, mainly total phenols (A), total flavonoids (B), total carotenoids (C), and ascorbic acid (D). Except for carotenoids, the other three bioactive substances showed decreasing trends after storage, but this decrease was not the same across different apricot varieties. The three apricot varieties with the highest total phenolic content in the initial state were SMO, SSN and AKI, and their total phenolic content decreased by 13.8%, 39.4% and 37.6%, respectively, after shelf life. The three apricot varieties with the lowest total phenolic content in the initial state were CZG, KTY and YU, whose total phenolic content decreased by 51.7%, 27.9% and 56.8%, respectively, after shelf life. The three apricot varieties with a higher initial total phenol content were found to have better tolerance for storage than fruits with lower initial total phenol content, and they also showed less decrease in total phenol content overall. Apricots are generally high in initial total flavonoid content because of the yellow color of the fruit skin and flesh. However, the flavonoid content of apricots decreased significantly after storage and shelf life. The apricots with the highest decrease in total flavonoid content after storage were YU, KTY, CZG and YBI, with a decrease of >60% overall. These four varieties of apricots also happened to be varieties with low storage tolerance. The trend of ascorbic acid was the same as that of total phenol and total flavonoid content. The decline in ascorbic acid in 21 apricot varieties after storage and shelf life ranged from 42.5% to 84.8%. The post-shelf life decrease in ascorbic acid in the HMI, JTG, KTY, SMO, SMI and SSN varieties was >70%, while the loss of ascorbic acid in the YU and LTI varieties was <50%. It was found that changes in ascorbic acid content and chilling injury occurrence and storage tolerance of apricots may not be correlated. From Figure 1, it can be seen that the apricots all exhibited color changes, which was mainly due to the accumulation of carotenoids. The total carotenoid content at the initial state was <0.06 mg·g^−1^. After 50 days of cold storage, total carotenoids of AKI, BTU, JTG, SMI, SSN and ZZU increased substantially and continued to increase after shelf life. Combined with the color of the apricots, it can be seen that these six apricot varieties are orange in color when ripe. The remaining apricot carotenoids, which appear yellow when ripe, do not accumulate as well as orange ones after storage. In addition, five apricot varieties, CZG, HMI, DJE, KTY and LSG, showed a decrease in carotenoid content after shelf life compared to the storage samples. All five apricots had chilling injuries, and except for LSG, were not storage tolerant.

### 3.4. Cluster Heat Map Analysis of 21 Apricot Varieties for Storage Tolerance

The above tests showed that the 21 varieties of apricots exhibited different physiological changes in terms of storage tolerance and chilling injury occurrence under various storage conditions, which also led to large differences in fruit storage tolerance. Therefore, this study carried out a standardized heat map analysis of data changes involving 18 physiological qualities of 21 varieties under three conditions: original samples, post-cold storage, and post-shelf life. The aim was to observe and analyze important factors that influence the greater variability across apricot varieties in storage tolerance.

As can be seen from Figure 7, the levels of MDA and DPPH, SOD, and O_2_^−^ resistance activities were low in various apricot varieties just after harvesting. By comparing changes in the standardized characteristic values of the physiological indicators of the original samples, the fruits could be classified into two categories based on storage tolerance. The first category demonstrated storage tolerance, including AKI, ZZU, SSN, KCI, LTI, BTU, LGG, DLI, SMI, and SMO. Each of these varieties could maintain good physiological quality during 50 d of cold storage and also ripen well during post-storage shelf life. This may be because these 10 varieties have high levels of ROS production and scavenging, which can maintain the low-temperature tolerance of the fruits. Usually, a prolonged cold storage environment stimulates the synthesis of large amounts of ROS such as O_2_^−^ and H_2_O_2_ in fruits. Stimulated by ROS, fruit cells of these 10 varieties can rapidly activate the activity of SOD, CAT and other enzymes that scavenge ROS, enhance antioxidant activity, and reduce toxic effects brought about by superoxide anions. Thus, this maintains a dynamic balance between the production and scavenging of ROS in apricots and reduces the chilling injury brought about by cold stress. The second category includes apricots that did not demonstrate storage tolerance, which include SJG, LSG, XBI, BSI, JTG, HMI, DJE, YU, KATY, CZG, and YBI. These 11 apricot varieties produced excess O_2_^−^ and H_2_O_2_ superoxide anions during low temperature stimulation, and the synthesis of cellular antioxidant active substances was slow. This caused disruptions in terms of ROS metabolic balance and damage to fruit cell membrane structures. Thus, these varieties were continuously damaged by cold stress during prolonged cold storage, resulting in more serious chilling injuries.

## 4. Discussion

### 4.1. The Metabolic Imbalance of Reactive Oxygen Species Leads to the Occurrence of Fruit Chilling Injury

When plants are subjected to external environmental stress, the metabolic activity of the plant produces a large number of changes to adapt to the external environment. Under normal conditions, bioactive substances and certain enzymes in the plant can scavenge these ROS that are produced in large quantities due to stress [13,29] However, when the level of ROS production exceeds the level that can be removed by the plant body, this causes excessive accumulation of ROS [30,31]. This has toxic effects on plant cells, affects cell membrane permeability, and disrupts the normal stress-resistance function of the fruit [32].

From our experimental results and data analysis, it was hypothesized that the important factors leading to chilling injury in apricot fruit mainly originated from an imbalance in ROS metabolism [33,34]. When postharvest fruits and vegetables are subjected to cold stress, ROS are produced and accumulate, causing oxidation chain reactions that lead to enzyme inactivation, cell membrane lipid peroxidation, protein degradation, and DNA damage. Ultimately, this leads to cell membrane oxidative damage and loss of cell membrane function [13,35] reported that the membrane lipid structure of mango pericarp cells is altered under cold stress. This leads to weakened function of proteins on the cell membrane, including a ROS scavenging system and the accumulation of oxidative damage substances such as ROS, resulting in damage to membrane structures and loss of protein function. When 11 storage intolerant varieties, such as YU, JTG and BSI, were stored at 0 °C for 50 d, the prolonged low-temperature environment stimulated the mitochondria of fruit cells to produce large amounts of superoxide anion radicals and ROS such as hydrogen peroxide, which are maintained at the ROS level for a long time. Additionally, with extended storage duration, the enzymatic activities of ROS scavenged by SOD and CAT and the contents of bioactive substances represented by total phenols and total flavonoids were also lower in the 11 varieties with a lack of storage tolerance such as YU, JTG and BSI, than in the 10 varieties with storage tolerance such as SSN and DLI. Since the scavenging of ROS by plant antioxidant systems consists mainly of enzymatic and non-enzymatic systems, all 10 varieties of apricots with storage tolerance had high SOD and CAT enzyme activities and elevated total phenol and total flavonoid contents [36]. In addition, this study also found that 11 varieties of apricots with higher initial sugar content and acidity were favorable for maintaining good physiological status under low-temperature conditions. The LTI variety had an initial sugar content of 135.51 mg·g^−1^ and SMO titratable acidity content of 18.72 mg·g^−1^. This may be because sugars and acids provide the necessary carbon skeletons for secondary metabolism in postharvest fruits and vegetables [37,38]. Therefore, apricots with higher sugar and acid content at harvest are also able to maintain more stable ROS production and scavenging functions during cold storage. In summary, our results indicated that prolonged low-temperature storage disrupts the original ROS balance inside the cells of 11 apricot varieties which exhibited a lack of storage intolerance. This results in continuous excessive accumulation of ROS, depletion of a large number of bioactive substances and reduced ROS scavenging enzyme activity, resulting in severe membrane lipid oxidation. This leads to the evident chilling injury phenomenon of peel-pitting spots, watery injuries inside or outside, and browning inside [39]. The two apricot varieties YBI and KATY had the most evidence of these outcomes, with excessive softening of apricots after cold storage, resulting in juice loss and severe browning, with YBI even reaching a 100% damage rate after storage. In addition to the chilling injury symptoms mentioned above, the apricots also exhibited collapse and the inability to turn ripe. This phenomenon of increased shelf life chilling injury and inability to ripen after prolonged cold storage is more evident in drupes such as peaches, apricots, and plums. A study by Zhao et al. [23] concerning nectarines subjected to low-temperature storage noted that although nectarines do not exhibit visible chilling injuries during low-temperature storage, rapid chilling injuries may occur during post-storage shelf life, accelerating nectarine decay. Therefore, when conducting storage tolerance and chilling injury studies on peach, apricot, and plum fruits, more attention needs to be paid to chilling injury symptoms during shelf life after cold storage.

### 4.2. Apricots with High Initial Sugar, Acid and Other Nutrients Content Are Resistant to Low-Temperature Storage

The results of cluster heat map analysis revealed that although all 21 apricot varieties came from high latitudes in northern China, there were very large differences in tolerance to chilling injury. In this study, clustering results were combined with the different geographical locations and initial physiological characteristics of 21 apricot varieties. Further analysis of the causes revealed that all the apricots with a lack of storage tolerance originated from northern China. As shown in Figure 8, except for YBI, the remaining nine varieties with a lack of storage tolerance, LSG, KATY, CZG, SJG, BSI, XBI, JTG, YU, DJE, and HMI, were all apricot varieties with hairy fruit surfaces. Additionally, 7 of the 10 storage-resistant apricot varieties were from northwest China. Except for SSN and BTU, the remaining eight apricots varieties had hairless fruit surfaces. Fan et al. [40] evaluated the sugar, acid, and phenolic contents of seven apricot varieties and found that fructose, sucrose, and malic acid levels of apricots from northwest China, such as SMI and AKU, were higher than those from north China. The higher sugar and acid contents may improve the storage properties of apricots, in addition to the sensory quality during consumption. Combined with the results of this study, our analysis suggested that the storage tolerance of apricots may be related to their origin and intrinsic characteristics, in addition to the initial bioactive substance content and the level of ROS [41]. Apricots from northwest China have generally higher storage tolerance. This may be attributed to the fact that apricots of the relevant varieties in the regions are grown and developed in an environment with high light intensity, low humidity, and large diurnal temperature differences for a prolonged period. These external environments may result in certain genes or proteins in apricots that can adapt to such stresses. SSN, SMI, DLI, AKI, LGG, and KCI represent apricots from northwest China that accumulate more sugars and acids to serve as substrates for respiratory metabolism and carbon skeletons for secondary metabolism, in response to possible stresses during growth due to diurnal temperature differences and rapid climate changes [42]. In addition, these apricots also accumulate more phenolic substances and bioactive substances such as ascorbic acid and maintain high levels of intercellular ROS metabolism [43], achieving rapid and stable resistance to stress. This enables the fruits of these varieties to maintain good storage tolerance characteristics and shelf performance after long periods of storage at low temperatures [44,45].

## 5. Conclusions

Our results showed that 10 fruit varieties, of which AKI, ZZU, BTU, LTI and SMO are representative, stimulated higher levels of antioxidant activities such as phenolics, flavonoids and POD in low-temperature environments, maintaining the dynamic balance of ROS production and scavenging. This enhances the resistance of the fruit to stress and improves storage tolerance. In addition, characteristics of the levels of tolerance during fruit growth were related to their place of origin. The 10 apricot varieties with chilling storage tolerance, of which SSN, SMI, and DLI were representative, were all from the northwestern region with diurnal temperature differences and rapid climate change. These fruits maintained high initial levels of energy required for vital activities such as sugar and acid metabolism, as well as the accumulation of substances such as flavonoids and phenolics. This higher stress tolerance helps stabilize the physiological state of the fruit during stress. In summary, combined with the geographic distribution of 21 varieties of apricots, this study illustrated the relationship between the incidence of chilling injury and ROS metabolism in 21 varieties of apricots and provides a theoretical basis for the postharvest storage and transportation methods and techniques involving apricots.

## Figures and Tables

**Figure 1 foods-12-02378-f001:**
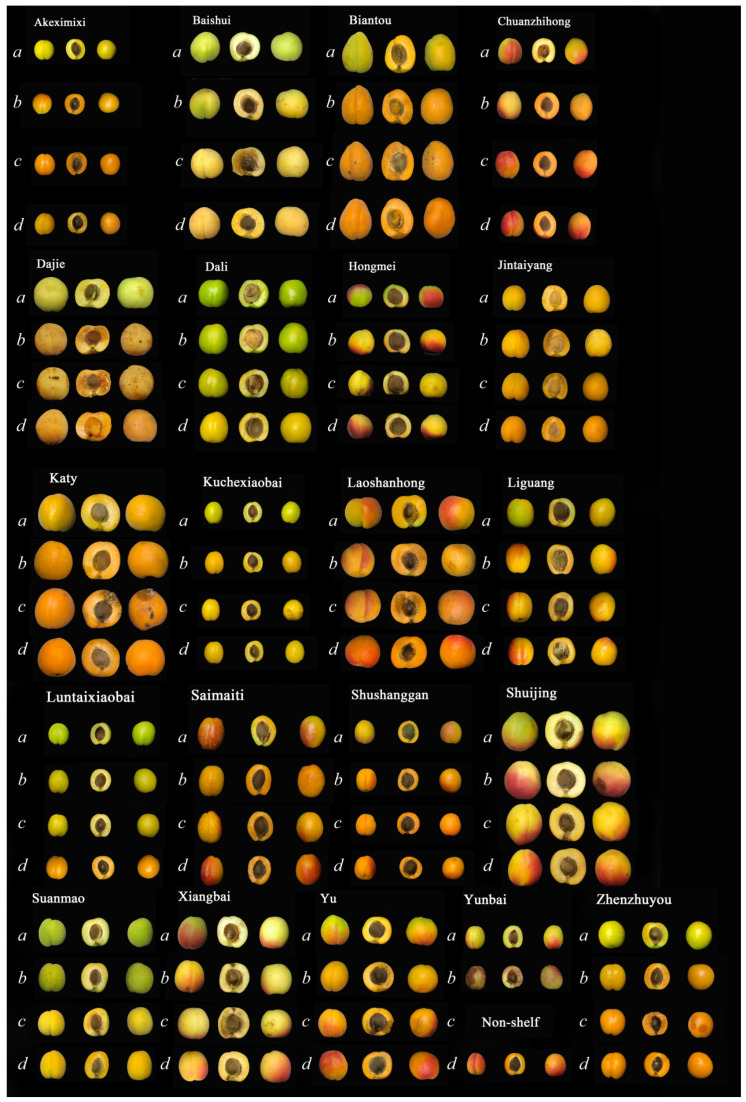
The changes in appearance of 21 apricots after low-temperature storage and shelf life. (Note: The italicized lowercase ***abcd*** marked in this figure represented the status of apricot samples at different maturity and storage periods. The ***a*** represented raw apricot fruits after harvest, ***b*** represented apricot fruits after storage for 50 d, ***c*** stands for apricot fruits after storage for 50 d and shelf life for 6–8 d, and ***d*** stands for naturally mature apricot fruits).

**Figure 2 foods-12-02378-f002:**
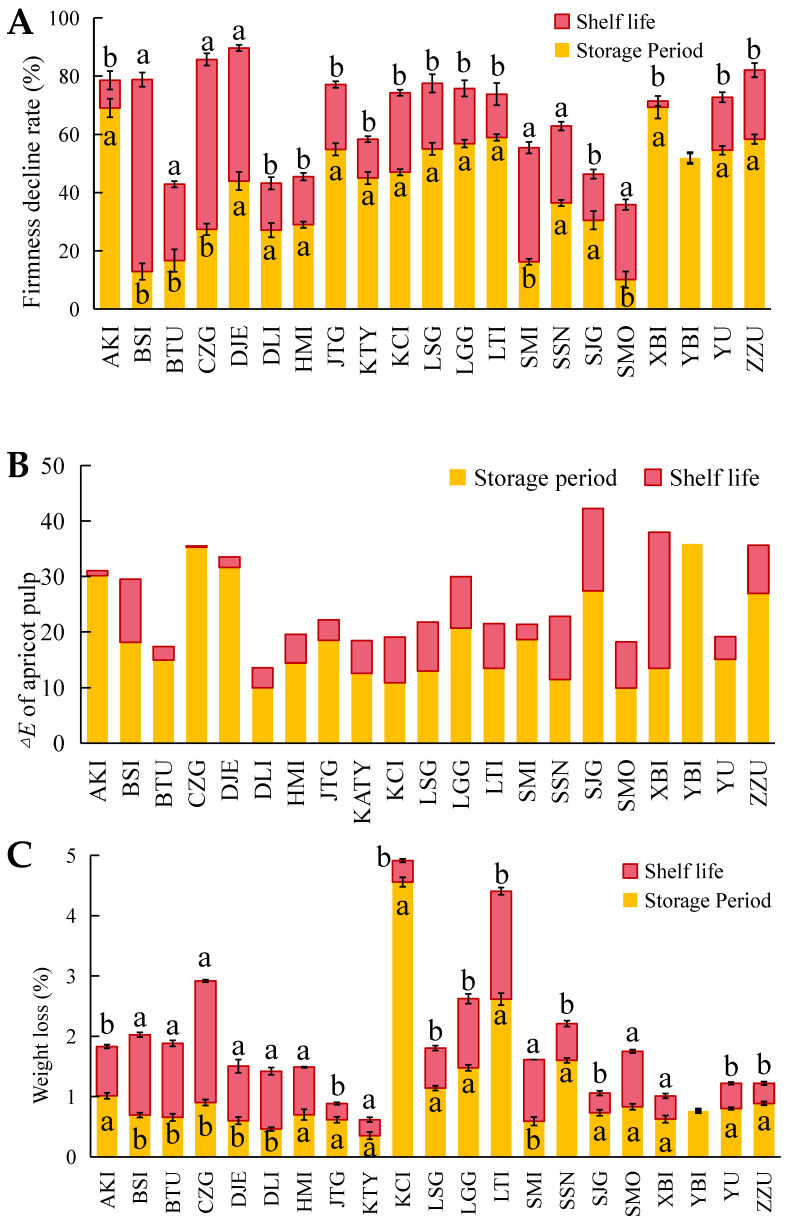
The changes of Firmness decline Rate (**A**), Δ*E* of apricot pulp (**B**) and Weight loss Rate (**C**) of 21 apricots after low-temperature storage and shelf life. (Note: this figure is a stacked bar chart. The red and yellow bars represent the index changes of apricots during storage and shelf life period, and the sum of red and yellow bars represents the total changes of apricots after storage and shelf life. The lowercase letters on the bar indicate the significant difference in value between storage and shelf life, and the different letters indicate the significant difference).

**Figure 3 foods-12-02378-f003:**
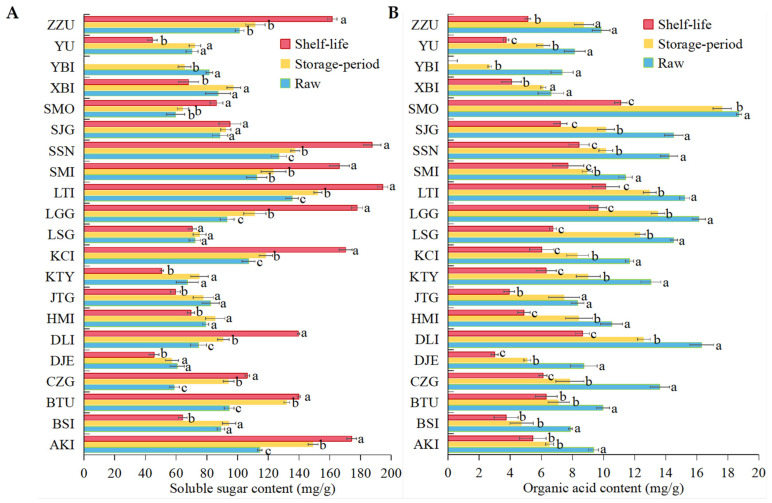
Changes in soluble sugar (**A**) and Organic acid content (**B**) in 21 apricots during initial, low-temperature storage and shelf life. (Note: The different lowercase letters on the bars in this figure represent the significance of difference. The significance of difference was the result of comparing the raw, storage and shelf life of apricots in the same variety. The same lowercase letters indicate no significant difference in the raw, storage and shelf life period of apricots in one same variety, while different lowercase letters indicate significant difference).

**Figure 4 foods-12-02378-f004:**
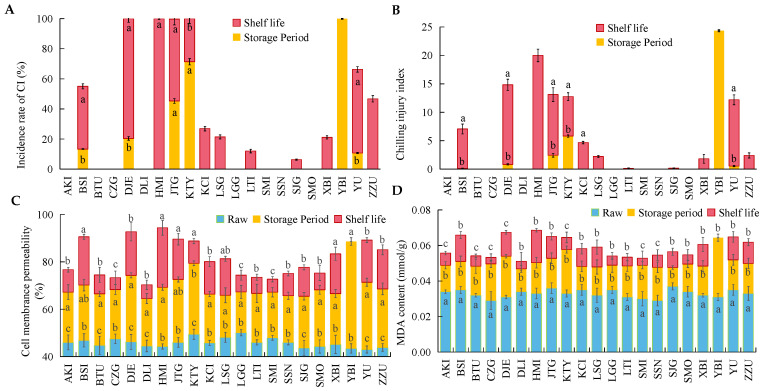
The changes in Chilling Injury Rate (**A**), Chilling Injury Index (**B**), Cell Membrane Permeability (**C**) and malondialdehyde (**D**) content of 21 apricot cultivars during cold storage and shelf life. (Note: this figure is a stacked bar chart. The red and yellow bars in (**A**,**B**) represent the index changes of apricots during storage and shelf life period, and the sum of red and yellow bars represents the total changes of apricots after storage and shelf life. The blue, red and yellow bars in (**C**,**D**) represent the index changes of apricots during raw, storage and shelf life period, and the sum of blue, red and yellow bars represents the sum of values of apricots among raw, storage and shelf life. The lowercase letters on the bar indicate the significant difference in value among raw, storage and shelf life, and the different letters indicate the significant difference).

**Figure 5 foods-12-02378-f005:**
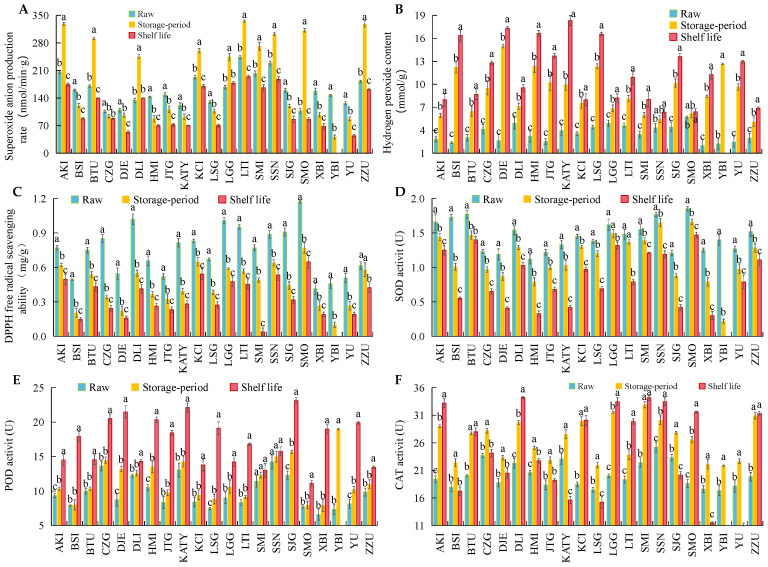
Changes of Superoxide Anion Production Rate (**A**), Hydrogen Peroxide Content (**B**), DPPH Free Radical Scavenging Capacity (**C**), SOD Activity (**D**), POD Activity (**E**) and CAT Activity (**F**) of 21 apricots after low-temperature storage and shelf life. (Note: The different lowercase letters on the bars in Figure 3 represent the significance of difference. The significance of difference was the result of comparing the raw, storage and shelf life of apricots in the same variety. The same lowercase letters indicate no significant difference in the raw, storage and shelf life period of apricots of in one same variety, while different lowercase letters indicate significant difference).

**Figure 6 foods-12-02378-f006:**
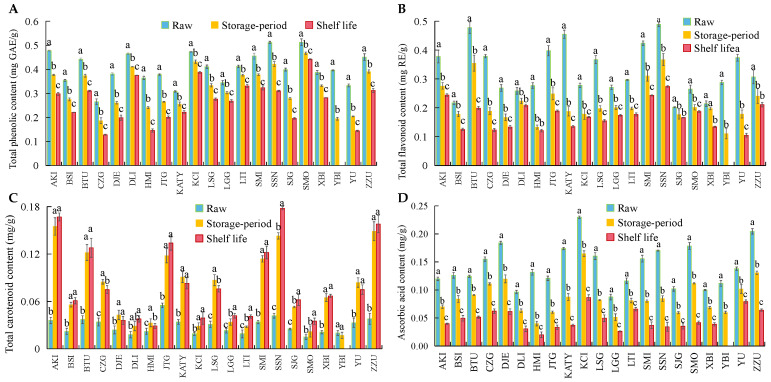
The changes of Total Phenols (**A**), Total Flavonoids (**B**), Total Carotenoids (**C**) and Ascorbic Acid (**D**) in 21 apricots during low-temperature storage and shelf life. (Note: The different lowercase letters on the bars in Figure 3 represent the significance of difference. The significance of difference was the result of comparing the raw, storage and shelf life of apricots in the same variety. The same lowercase letters indicate no significant difference in the raw, storage and shelf life period of apricots of in one same variety, while different lowercase letters indicate significant difference).

**Figure 7 foods-12-02378-f007:**
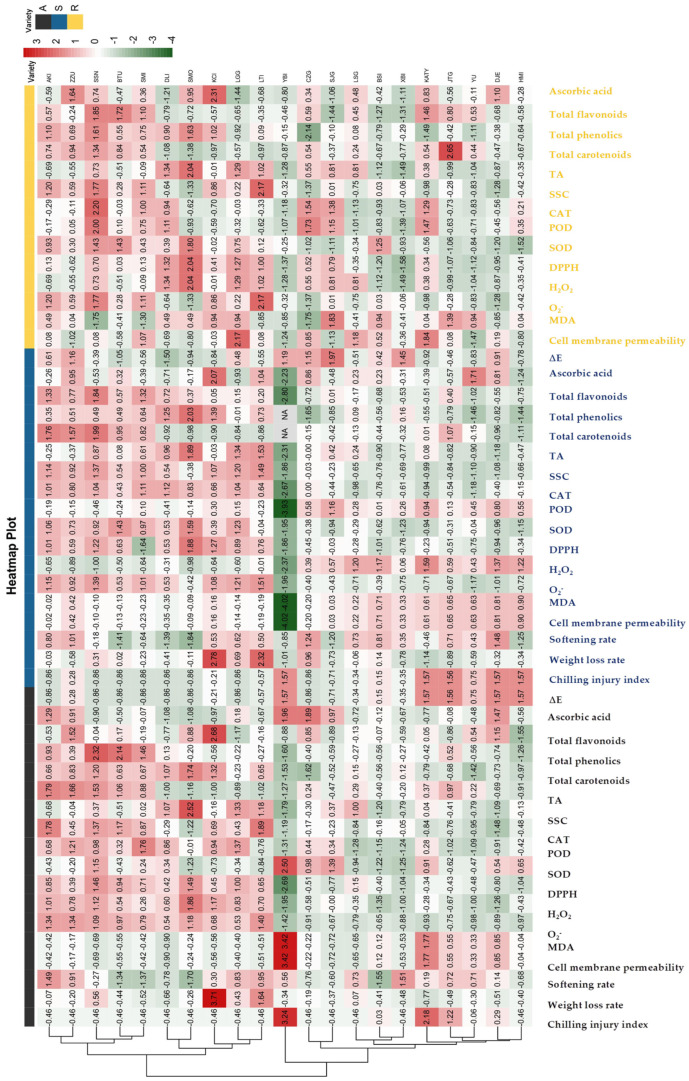
The standardized cluster heat map of 21 kinds of apricot origin, low temperature storage and shelf physiological index changes after storage.

**Figure 8 foods-12-02378-f008:**
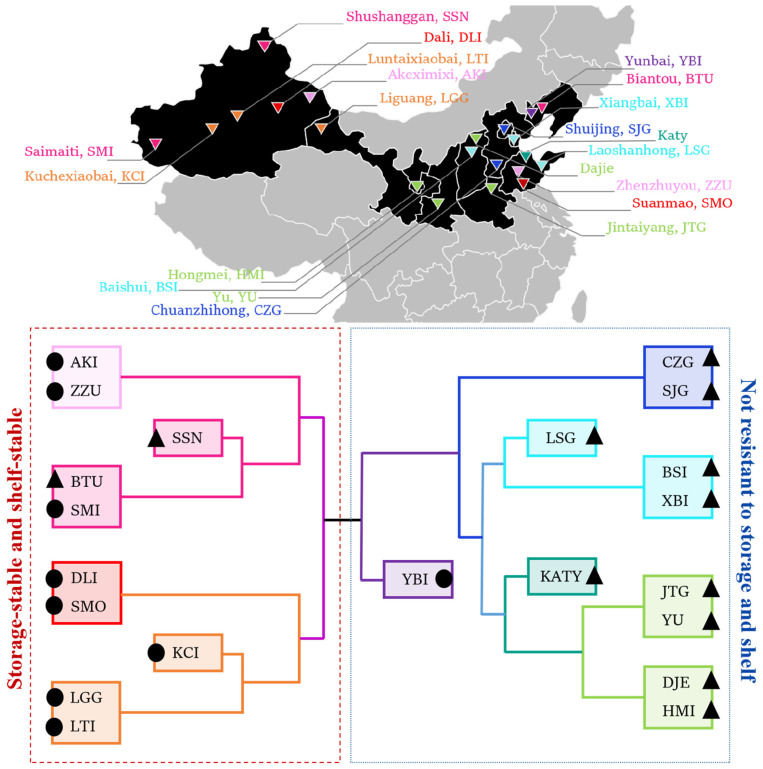
Classification and distribution of storage tolerance of 21 apricot species. (Note: The triangle and round symbols marked on the apricot fruit variety abbreviations indicate whether the apricot fruit has a smooth surface or a hairy surface. The round symbol indicates the apricot fruit variety with a smooth surface, and the triangular symbol indicates an apricot fruit variety with a hairy surface).

**Table 1 foods-12-02378-t001:** Harvest time of 21 apricot varieties at green mature and full mature stages.

Apricot	Abbreviation	Harvest Time	Surface	Producing Area	Apricot	Abbreviation	Harvest Time	Surface	Producing Area
Akeximixi	AKI	28th May	Hairless	Xinjiang	Baishui	BSI	19th May	Hairy	Shanxi
Biantou	BTU	15th June	Hairy	Liaoning	Chuanzhihong	CZG	18th June	Hairy	Hebei
Dajie	DJE	22nd May	Hairy	Shanxi	Dali	DLI	2nd July	Hairless	Xinjiang
Hongmei	HMI	17th June	Hairy	Ningxia	Jintaiyang	JTG	15th June	Hairy	Henan
Katy	KATY	20th May	Hairy	Shandong	Kuchexiaobai	KCI	6th June	Hairless	Xinjiang
Laoshanhong	LSG	25th May	Hairy	Shandong	Liguang	LGG	10th June	Hairless	Gansu
Luntaibai	LTI	4th June	Hairless	Xinjiang	Saimaiti	SMI	20th June	Hairless	Xinjiang
Shushanggan	SSN	18th June	Hairy	Xinjiang	Shuijing	SJG	12th June	Hairy	Beijing
Suanmao	SMO	18th June	Hairy	Shandong	Xiangbai	XBI	20th June	Hairy	Tianjin
Yunbai	YBI	5th July	Hairless	Liaoning	Yu	YU	5th June	Hairy	Shaanxi
Zhenzhuyou	ZZU	10th June	Hairless	Shandong					

## Data Availability

The data used to support the findings of this study can be made available by the corresponding author upon request.

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
