# Peer review of "Effects of Reactive Oxygen Levels on Chilling Injury and Storability in 21 Apricot Varieties from Different Production Areas in China"

_foods, 2023, doi:10.3390/foods12122378_

Round 1
Reviewer 1 Report
I am honored by opportunity to review manuscript “Effects of reactive oxygen levels on chilling injury and storability in 21 apricot varieties from different production areas in China” by Qi Xin et al.
The main strength of manuscript is comparative view on cold storage and shelf life of 21 varieties of apricot in terms of weight loss, color change, firmness, soluble sugar and organic acid, chilling injury rate, chilling injury index, cell membrane permeability, malondialdehyde, superoxide anion production rate, hydrogen peroxide content, DPPH, SOD, POD Activity, CAT Activity, total phenols, total flavonoids, total carotenoids and ascorbic acid. Such comprehensive presentation of valuable results and varietal diversity can full lack of results related to apricot storage.
Despite abundance of presented data, manuscript suffers from many deficiencies, primarily in data presentation, noncompliance to manuscript preparation (wrong citation style), English language, method description, lack of literature citation, lack of presentation of statistical results… All of these deficiencies must be addressed.
Although plethora of results are presented, authors also uses results which is not presented (ΔE of peel, color values, presence or absence of hair of apricot…)
Main deficiency of presented paper is discussion of results. In discussion authors tried to explain three points:
· storability of 21 variety of apricots,
· relation of ROS to chilling injury, storage and shelf life and
· relation of storage to geographical origin and external appearance of apricots.
Currently, discussion is very hard to follow, so I suggest authors to divide discussion to subsection out of which later conclusions will be made.
The last remark is text which provide information, but literature is not cited (for example, lines 49-60; 70-74; 79-89, 90-92, 462-471, 477-479, 482-485) should be either deleted or supported by literature.
When all changes are made, please make necessary corrections to abstract and conclusion, according to instructions for authors for Foods journal.
I encourage authors to make all necessary changes and to resubmit since data provided is very valuable, having in mind global decline in apricot production. I personally very much appreciate results of flesh firmness, since this is one of the main problems of apricot storage.
Specific comments:
INTRODUCTION
Line 51 “China has 90 % of the world's apricot germplasm resources” can you backup this with literature?
Lines 53-56 “Although rich in germplasm resources, the apricot consumption and processing industry in China has gradually shrunk in recent years mainly due to the lack of storage and preservation techniques, resulting in stagnant consumption” – this also happens around the world. Can you find some literature to backup this statement?
Line 57-60 this sentence should go at the beginning of last paragraph (aim of the work, please look at the instructions for authors at https://www.mdpi.com/journal/foods/instructions)
Line 69 “shelving” is very unusual term. Please use “shelf life” throughout text.
Lines 102-107. Please combine lines 53-56 and 102-107 into separate paragraph at the end of introduction which will define aim of the study. Keep in mind that title, aim title, method, results, discussion and conclusion should be aligned!
MATERIALS AND METHODS
Lines 109 – 134. Process of transport and storage of apricots MUST be expalind in more detail! By reading text now it seems that apricots were transported from field at 0 °C (Line 114) than heated up to 10 °C (line 123) and then again cooled to 0 ± 1 ℃ (this heating cooling procedure will affect storage of apricot)! Second question is related to fruit put immediately at 25 °C for 6-8 days (lines 125-127) – where are results of those fruits? Third question is – how long was shelf life lasted (lines 131-134)? Please avoid using “shelving” instead use “shelf life”. Please rewrite this chapter so that all points of taking samples are clearly stated.
Formulae 1 – please make correction to “Weoghless rate” (wightloss?)
Line 141 – precision of analytical balance is usually presented as 0.0001 of a gram.
Line 144 “The color parameters L*, a* and b* of peel and pulp” – there is no results of peel! Please make correction
lines 152-156. Please rewrite chapter about carotenoids determination. In paper Barba, Hurtado, Mata, Ruiz, and de Tejada (2006) out of 6 solvents hexane/acetone/ethanol (50:25:25 v/v/v) showed the best results. Please state which solvent did you use and briefly explain method.
Line 163 “hand-held sugar meter” please state made and model for hand-held sugar met and all other critical equipment (spectrophotometer, balance, centrifuge, etc)
Line 237 – please use English!
Line 252 – ANOVA and Duncan multiple comparison method are used to determine significance of differences between apricots, however there is no presentation of ANOVA and Duncan differences within figures. Please find a way to present significance of differences between apricots.
RESULTS
In all figures error bars are presented. Are those error bars presents standard devations?
Lines 258-272 whole paragraph explains changes of colors (lines 264-265 “changed from white and light yellow to orange), however ΔE presents only difference in color. I order to support paragraph (lines 258-272) results L*, a*, b* or chroma (C*) must be presented.
Line 263, there is no results of ΔE of apricot skin.
Figure 1. Please define a, b, s, and d for figure 1A. Also, figure 1A MUST be clearer, from presented picture it is almost impossible to see differences. Figures 1B, C and D – please explain how shelf life loss was calculated. As mentioned earlier, please find a way to present results of statistical analysis (for example is there difference in firmness loss between KCI, LSG, LGG and LTI).
Line 281. “Figure 1B, none of the 21 apricot varieties had a quality loss rate” on 1B weight loss is presented (not quality loss)!
Line 238 On figure 1B weight loss is presented (not quality loss)!
Line 286 “Combining Fig. 1A and Fig. 2A” there is almost impossible to drow any conclusion from picture 1A. Picture MUST be improved!
Figure 2 – all bars are tiny it is very difficult to see differences. Please find a way to present data more clear.
Line 310 – “As can be seen from Figure 1A” please improve figure 1A. Apricot fruit degradation is not uniformly present so far. Your data (figure 1A) could shed light on types of chilling injury in apricot fruit (“water staining”, browning of the flesh observed punctiform depressions of the epidermis). Please make supplementary material in which all types of chilling injury will be presented.
Line 366 “>”?
Figure 4 – all bars are tiny it is very difficult to see differences. Please find a way to present data more clear.
Figure 5 – all bars are tiny it is very difficult to see differences. Please find a way to present data more clear.
DISCUSSION
Lines 462-472 is common knowledge. Please make it shorter and find appropriate citations.
Figure 7 is not cited in text of manuscript. Figure legend must be added and all symbols (circles, triangles…) must be explained.
Lines 528 and 530 – presence or absence of hairy fruit surfaces – it is very unusual to present new data in discussion (description of varieties, including presence of absence of hair should be presented in material and methods or results chapter).
Lines 542-544. Please be careful with conclusions. Maybe it is just coincidence and that varieties SSN, SMI, DLI, AKI, LGG, and KCI are high in sugar and acids and that has nothing to do with geographical origin. The only way to support your clam is to establish experiment with those varieties in different parts in China and to see weather sugar content will be different.
CONCLUSION
Lines 552-556 Conclusion should be related to results and discussion. Please delete sentence.
English MUST be improved. Shelving MUST be excluded.
Author Response
Response Letter
Dear editor:
Thank you very much for giving us an opportunity to revise our manuscript entitled “Effects of reactive oxygen levels on chilling injury and storability in 21 apricot varieties from different production areas in China” (ID: foods-2418778).
We have revised our manuscript carefully according to the comments, and would like to re-submit it for your consideration. The amendments are highlighted in red font in the revised manuscript. Our responses to the referees are listed by point to point in the following sections. If there are any deficiencies in this manuscript modification, we will fully cooperate to editor and revise it again in the next time.
In addition, we request the editor to rearrange our manuscript to make Figure 1 larger so that readers can better observe the color changes and chilling damage of apricot fruits of different varieties after storage and shelf life.
We would like to express our great appreciation to you and reviewers for the comments and hope that revised manuscript is acceptable.
Look forward to hearing from you.
Yours sincerely,
Prof. Bangdi Liu
Academy of Agricultural Planning and Engineering, Ministry of Agriculture and Rural Affairs of the People's Republic of China
PO Box 111, No. 41 Maizidian Road, Beijing, 100083, China
E-mail: [email protected]
Reply to Referees
to Referees 1
Question1:
Line 51 “China has 90 % of the world's apricot germplasm resources” can you backup this with literature?
Response:
Thanks for the reviewer's suggestion. We have modified this part and added related reference.
Question2:
Lines 53-56 “Although rich in germplasm resources, the apricot consumption and processing industry in China has gradually shrunk in recent years mainly due to the lack of storage and preservation techniques, resulting in stagnant consumption” - this also happens around the world. Can you find some literature to backup this statement?
Response:
Thanks for the reviewer's suggestion. We have modified this part and added related reference.
Question3:
Line 57-60 this sentence should go at the beginning of last paragraph (aim of the work, please look at the instructions for authors at https://www.mdpi.com/journal/foods/instructions)
Response:
Thanks to the reviewer's suggestion. We have modified this place and added the above sentence to the end as suggested.
Question4:
Line 69 “shelving” is very unusual term. Please use “shelf life” throughout text.
Response:
Thanks for the reviewer's suggestion. We have checked the full manuscript and replaced all "shelving" with "shelf life" as suggested.
Question5:
Lines 102-107. Please combine lines 53-56 and 102-107 into separate paragraph at the end of introduction which will define aim of the study. Keep in mind that title, aim title, method, results, discussion and conclusion should be aligned!
Response:
Thanks to the suggestions of the reviewers, we have merged line53-56 from the abstract into the introduction, and have organized the section stating the research purpose of this paper into a separate paragraph, ensuring that our objectives are aligned.
Question6:
Lines 109 – 134. Process of transport and storage of apricots MUST be expalind in more detail! By reading text now it seems that apricots were transported from field at 0 °C (Line 114) than heated up to 10 °C (line 123) and then again cooled to 0 ± 1 ℃ (this heating cooling procedure will affect storage of apricot)! Second question is related to fruit put immediately at 25 °C for 6-8 days (lines 125-127) – where are results of those fruits? Third question is – how long was shelf life lasted (lines 131-134)? Please avoid using “shelving” instead use “shelf life”. Please rewrite this chapter so that all points of taking samples are clearly stated.
Response:
Thanks for the reviewer's very professional and accurate suggestions. We have revised all the three suggestions and will answer them one by one
First, we have rewritten parts 1.1 and 1.2 according to the reviewer's suggestion, so that readers can better understand how we collected 21 kinds of apricot fruits and processed them. In addition, the original reference to transport at 0℃ followed by 10℃ standing was completely caused by our clerical error. We have revised the word description to transport at 10℃ followed by 10℃ standing.
Second, this result was not chosen to be presented in the paper, because this result is not important to the conclusion of the paper, so we deleted this sentence.
Third, we have replaced "shelving" into "shelf life" in the full text, and rewritten the way of shelves and sample collection.
Question7:
Formulae 1-please make correction to “Weoghless rate” (wightloss?)
Response:
Thanks for the reviewer's suggestion. We have modified this part. Due to a clerical error, the spelling was wrong. We have checked the full manuscript and corrected it.
Question8:
Line 141-precision of analytical balance is usually presented as 0.0001 of a gram.
Response:
Thanks for the reviewer's suggestion. We have corrected the clerical error in Line141.
Question9:
Line 144 “The color parameters L*, a* and b* of peel and pulp” – there is no results of peel! Please make correction
Response:
Thanks for the reviewer's suggestions, we have modified according to the reviewer's suggestions. The color parameters related to peel did not appear in the paper. We have modified this sentence and checked the description of color in the whole paper to avoid the description of color of peel.
Question10:
lines 152-156. Please rewrite chapter about carotenoids determination. In paper Barba, Hurtado, Mata, Ruiz, and de Tejada (2006) out of 6 solvents hexane/acetone/ethanol (50:25:25 v/v/v) showed the best results. Please state which solvent did you use and briefly explain method.
Response:
Thanks to the reviewer's very professional and accurate suggestion. We have rewritten the part of carotenoid determination and described the solvent detection system in detail.
Question11:
Line 163 “hand-held sugar meter” please state made and model for hand-held sugar met and all other critical equipment (spectrophotometer, balance, centrifuge, etc)
Response:
Thanks to the reviewer's suggestion. We have added the instrument model and origin used in the manuscript.
Question12:
Line 237 – please use English!
Response:
Thanks for the reviewer's suggestion. We have removed this formula because it is a clerical error and is not needed.
Question13:
Line 252 – ANOVA and Duncan multiple comparison method are used to determine significance of differences between apricots, however there is no presentation of ANOVA and Duncan differences within figures. Please find a way to present significance of differences between apricots.
Response:
Thanks for the reviewer's suggestion. We have verified all the figures and added significance of differences. Lowercase letters represent differences between groups. Add a note of difference description under the corresponding figure.
Question15:
In all figures error bars are presented. Are those error bars presents standard devations?
Response:
Thanks for the reviewer's suggestion. The error bar represents the standard error of the test, and we have supplemented the explanation in the method part. In addition, to express the difference more clearly, lowercase letters have been used in the picture to show the difference.
Question16:
Lines 258-272 whole paragraph explains changes of colors (lines 264-265 “changed from white and light yellow to orange), however ΔE presents only difference in color. I order to support paragraph (lines 258-272) results L*, a*, b* or chroma (C*) must be presented.
Response:
Thanks for the reviewer's suggestions. We think the reviewer's suggestions are very professional, but we don't think L*, a*, b* or chroma (C*) is needed here. This is because Fig 1A has been able to clearly show the overall color transformation of apricot fruits, and we believe readers can get relevant results from the intuitive photos.In addition, we did not mention the relevant values of L*, a*, b* or chroma (C*) in the whole paragraph of 258-272, so we think it is not necessary to add such data here. In addition, we have modified this paragraph and Figure 1A, especially adding the explanation of Figure 1A and the annotation of Figure 1A, so that readers can quickly compare the content of this paragraph with Figure 1A.
Although we added L*, a*, b* value data in a certain version of the paper, we found that this part of data could not bring much effective discussion and analysis to the paper, and the paper data was very miscellaneous, so we chose not to present L*, a*, b* or chroma (C*). However, if the reviewer and editor still think that this part of data needs to be added after this revision, we will modify it.
Question17:
Line 263, there is no results of ΔE of apricot skin.
Response:
Thanks for the reviewer's comments, we have modified here in the paragraph. Line263 does not correspond to apricot fruit color difference in FIG. 1C, but the overall appearance change in FIG. 1A. We have annotated and explained the marks in Figure 1A below the picture title, and elaborated Figure 1A near Line263. We believe that the addition of such information can more effectively help readers understand the content described in this paragraph.
Question18:
Figure 1. Please define a, b, s, and d for figure 1A. Also, figure 1A MUST be clearer, from presented picture it is almost impossible to see differences. Figures 1B, C and D – please explain how shelf life loss was calculated. As mentioned earlier, please find a way to present results of statistical analysis (for example is there difference in firmness loss between KCI, LSG, LGG and LTI).
Response:
Thanks for the reviewer's comments, we have modified the parts proposed by the reviewer. Due to our oversight, we did not add a note under Figure 1. We have annotated the lowercase italics abcd in Figure 1A, which represent the different states of the apricot fruit.
In addition, we have separated Figure 1A and Figure 1BCD so that the editor can display Figure 1A in a larger and clearer way. We have provided the HD original image of Fig 1A to the editor. We kindly ask the editor to rearrange the layout for us to show the content in Fig1A more clearly.
Thirdly, the measurement and calculation methods of quality loss, color difference and hardness loss ratio of apricot fruits under shelf life condition are exactly the same as those of fruits in storage period. There is no special detection and calculation method. However, we have added response description to the part of the method to avoid the situation that readers cannot understand.
Finally, we have added error line notes to all charts so that readers can better distinguish differences between groups.
Question19:
Line 281. “Figure 1B, none of the 21 apricot varieties had a quality loss rate” on 1B weight loss is presented (not quality loss)!
Response:
Thanks for the reviewer's suggestion. We have modified this part.
Question20:
Line 238 On figure 1B weight loss is presented (not quality loss)!
Response:
Thanks for the reviewer's suggestion. We have modified this part.
Question21:
Line 286 “Combining Fig. 1A and Fig. 2A” there is almost impossible to drow any conclusion from picture 1A. Picture MUST be improved!
Response:
Thanks for the reviewer's suggestion. We have modified this part. Fig. 1 has been split, and the photos of changes in the appearance of apricot fruits have been separately split as FIG. 1 for readers' clearer observation, and letters have been added to the bar chart to indicate significant differences between groups.
In addition, we request the editor to rearrange our manuscript to make Figure 1 larger so that readers can better observe the color changes and chilling damage of apricot fruits of different varieties after storage and shelf life.
Question22:
Figure 2 – all bars are tiny it is very difficult to see differences. Please find a way to present data more clear.
Response:
Thanks for the reviewer's suggestion. We have modified this part. We have enlarged the data bars in all figures and changed the colours of the different groups of bars so that readers can see the data more clearly.
Question23:
Line 310 -“As can be seen from Figure 1A” please improve figure 1A. Apricot fruit degradation is not uniformly present so far. Your data (figure 1A) could shed light on types of chilling injury in apricot fruit (“water staining”, browning of the flesh observed punctiform depressions of the epidermis). Please make supplementary material in which all types of chilling injury will be presented.
Response:
Thanks for the reviewer's suggestion. We have modified here. We separate the photo of apricot fruit appearance change (FIG. 1A) and provide a high-definition picture of FIG. 1A. We request the editor to enlarge FIG. 1A when typesetting. We are sure that different chilling injury in apricot fruit can be clearly observed in our high-resolution photos in new Fig 1.
Question24:
Line 366 “>”?
Response:
Thanks for the reviewer's suggestion. We have modified this part.
Question25:
Figure 4 – all bars are tiny it is very difficult to see differences. Please find a way to present data more clear.
Response:
Thanks for the reviewer's suggestion. We have modified this part. We have enlarged the data bars in all figures and changed the colours of the different groups of bars so that readers can see the data more clearly.
Question26:
Figure 5 – all bars are tiny it is very difficult to see differences. Please find a way to present data more clear.
Response:
Thanks for the reviewer's suggestion. We have modified this part. We have enlarged the data bars in all figures and changed the colours of the different groups of bars so that readers can see the data more clearly.
Question27:
Lines 462-472 is common knowledge. Please make it shorter and find appropriate citations.
Response:
Thanks to the reviewers for their comments, which we have revised here. The Lines 462-472 has been deleted and relevant literature added.
Question28:
Figure 7 is not cited in text of manuscript. Figure legend must be added and all symbols (circles, triangles…) must be explained.
Response:
Thanks for the reviewer's suggestion. We have modified here. Lines 462-472 have been cut. We also cited Fig. 7 in the corresponding position.
Question29:
Lines 528 and 530 – presence or absence of hairy fruit surfaces – it is very unusual to present new data in discussion (description of varieties, including presence of absence of hair should be presented in material and methods or results chapter).
Response:
Thanks for the reviewer's professional advice, we have made corresponding modifications. Indeed we found that the presence or absence of hairy fruit surfaces did not appear in the previous section, which was a great oversight on our article writing. We have added the surface and origin of apricot fruits in Table 1, which introduces all apricot fruits, in the methods part of the paper.
Question30:
Lines 542-544. Please be careful with conclusions. Maybe it is just coincidence and that varieties SSN, SMI, DLI, AKI, LGG, and KCI are high in sugar and acids and that has nothing to do with geographical origin. The only way to support your clam is to establish experiment with those varieties in different parts in China and to see weather sugar content will be different.
Response:
Thanks for the reviewer's very professional advice. We agree with the suggestions of the reviewer for the follow-up experiment, but we still want to reserve the content in this discussion part. Because, this is the discussion part, we can make reasonable inferences. We believe that we can deduce from the available data that apricot fruits from northwest China are relatively resistant to chilling injury and storage due to their high sugar and acid content. In the research on fruit storage in northwest China, some papers have proposed that, due to the larger climate temperature difference and higher geographical altitude in Xinjiang and Gansu, fruits in this region are more resistant to storage than those in the eastern region with lower altitude and smaller temperature difference. Therefore, the researchers concluded that the harsher growing environment may have caused a kind of long-term environmental stress on the plants, resulting in long-term acclimatisation to withstand the harsh environment.
Question31:
Lines 552-556 Conclusion should be related to results and discussion. Please delete sentence.
Response:
Thanks for the reviewer's suggestion. We have modified this place. We have shortened the Conclusion of Lines 552-556.

Reviewer 2 Report
Overall assessment: Interesting work, the planning of the research, conducting the experiments, as well as the obtained results do not raise any objections. Nevertheless, I have a few suggestions for you: in my opinion, many of the statements in the text, both in the introduction and in the discussion of the results, are not supported by quotations - please complete. In the Introduction section, you introduced some abbreviations - please explain them when using it for the first time and it would be good to use these abbreviations consistently later in the manuscript.
The presentation of the results is poor quality, especially Fogure 6 is very low quality. It can be done more professionally, needs to be improved.

Author Response
Response Letter
Dear editor:
Thank you very much for giving us an opportunity to revise our manuscript entitled “Effects of reactive oxygen levels on chilling injury and storability in 21 apricot varieties from different production areas in China” (ID: foods-2418778).
We have revised our manuscript carefully according to the comments, and would like to re-submit it for your consideration. The amendments are highlighted in red font in the revised manuscript. Our responses to the referees are listed by point to point in the following sections. If there are any deficiencies in this manuscript modification, we will fully cooperate to editor and revise it again in the next time.
In addition, we request the editor to rearrange our manuscript to make Figure 1 larger so that readers can better observe the color changes and chilling damage of apricot fruits of different varieties after storage and shelf life.
We would like to express our great appreciation to you and reviewers for the comments and hope that revised manuscript is acceptable.
Look forward to hearing from you.
Yours sincerely,
Prof. Bangdi Liu
中华人民共和国农业农村部农业规划与工程学院
麦子店路111号邮政信箱41号, 北京, 100083, 中国
电子邮件: [email protected]
to Referees 2
Question1:
Add a space between the number and the unit, such as “0 °C”.
Response:
Thanks for the reviewer's suggestion. We have checked the full manuscript and revised it as required.
问题2:
第 237 行中的“此处键入公式”是什么意思?需要什么声音公式?
响应:
感谢审稿人的建议。我们已经修改了这个地方。这是一个错误的加法公式,已被删除。
Question3:
please provide a more suitable description differences between date?
Response:
Thanks for the reviewer's suggestion. We have added lowercase letters to all the Figures to represent the significance of the difference.
Question4:
please rephrase sentence so that it is not similar to the sentences in the introduction and disscusion.
Response:
Thanks for the reviewer's suggestion. We have checked the introduction and discussion section and deleted some inappropriate discussion and description..

Reviewer 3 Report

Language expression should be moderately improved.
Author Response
to Referees 3
Question1:
Provide references.
Response:
Thanks for the reviewer's suggestion. According to the requirements, relevant references have been added to the marked part of the paper as support.
Question2:
Complete the full name before the first abbreviation.
Response:
Thanks for the reviewer's suggestion. We have checked the full manuscript and revised it as required.
Question3:
please provide a more suitable description differences between date?
Response:
Thanks for the reviewer's suggestion. We have added lowercase letters to all the Figures to represent the significance of the difference.
Question4:
please rephrase sentence so that it is not similar to the sentences in the introduction and disscusion.
Response:
Thanks for the reviewer's suggestion. We have checked the introduction and discussion section and deleted some inappropriate discussion and description..

Round 2
Reviewer 1 Report
Congratulations to authors! Significant improvement of paper has been made.
However, there is correction that must be done and it relates to:
· STATISTICS!, or to be more precise – how results of statistical evaluation is presented.
As someone who really needed paper, as you are going to publish now, to cite in terms of apricot variety differences, I must point out problem with statistic evaluation.
What a, b and c represents in figure 2 and 4? For example, it seems that a and b are difference in firmness decline between storage period and shelf life (although similarity between firmness loss in DJE (Fig 2A) which is around 50% raise question even to that statement.
I expected calculation of difference between varieties in terms of firmness decline separately for storage period, shelf life and storage period + shelf life. I found very interesting result between AKI and BSI (majority for firmness decline for AKI is during storage period, while BSI lost majority of firmness during shelf life), however, according to statistics there is no significant difference (or yellow parts are b, while red decline is b)! Please improve!
Regarding legend (explanation of yellow and red part, figure 2 and 4) – yellow is cold storage period? Red is shelf life? Yellow+red = total storage? Please improve.
Figures 3, 5 and 6. Please state on what lettering is related to? For example, if one compare only results within one variety (it must be since there is difference on soluble sugar content between ZZU c and DJE c (fig 3A)). This explanation should be stated in caption of figure. Explanation of lettering should be stated in figures where results of statistical data processing are presented.
--
Author Response
to Referees 1
Question1: STATISTICS!, or to be more precise-how results of statistical evaluation is presented. As someone who really needed paper, as you are going to publish now, to cite in terms of apricot variety differences, I must point out problem with statistic evaluation.
Response: Thank the reviewers for their comments. We have recalculated and substantially modified the data statistics in this revision. We have redrawn the Figures 2 to 6 so that readers can get useful information when they read the images.
Question2: What a, b and c represents in figure 2 and 4? For example, it seems that a and b are difference in firmness decline between storage period and shelf life (although similarity between firmness loss in DJE (Fig 2A) which is around 50% raise question even to that statement.
Response: Thanks to the reviewer's suggestions, we have modified Figure 2 to Figure 6 and added notes below the Figures title. The corresponding meaning of the lowercase letter abc in the figure is expounded in detail. In addition, since previous significance calculations tend to lead to comprehension problems, we have recalculated significance to avoid related issues raised by reviewers.
Question3: I expected calculation of difference between varieties in terms of firmness decline separately for storage period, shelf life and storage period + shelf life. I found very interesting result between AKI and BSI (majority for firmness decline for AKI is during storage period, while BSI lost majority of firmness during shelf life), however, according to statistics there is no significant difference (or yellow parts are b, while red decline is b)! Please improve!
Response: Thanks for the reviewer's suggestion, we have modified it. The significance of the difference in the revised figures 2 and 4 is compared with the values of storage and shelf life period.
Question4: Regarding legend (explanation of yellow and red part, figure 2 and 4) – yellow is cold storage period? Red is shelf life? Yellow+red = total storage? Please improve.
Response: Thanks for the reviewer's suggestions, we have modified this place. Since we did not provide detailed notes before, the reviewers were unable to understand the red and yellow bars in Figures 2 and 4 and their combined meaning. We have added a detailed description of this in the revised manuscript under the picture questions in Figures 2 and 4.
We added notes below: The Fig. 2. is a stacked bar chart. The red and yellow bars represent the index changes of apricots during storage and shelf life period, and the sum of red and yellow bars represents the total changes of apricots after storage and shelf life.
Question5: Figures 3, 5 and 6. Please state on what lettering is related to? For example, if one compare only results within one variety (it must be since there is difference on soluble sugar content between ZZU c and DJE c (fig 3A)). This explanation should be stated in caption of figure. Explanation of lettering should be stated in figures where results of statistical data processing are presented.
Response: Thanks to the reviewer's suggestion, we have modified this place. The lower-case letters in figures 3, 5, and 6 do indicate the difference between the values of apricot fruits of the same variety at different storage periods. We have added this detailed notes below the figure in Figure 3, Figure 5, and Figure 6 for ease of understanding.
